# Native Aortic Root Thrombosis in Hypoplastic Left Heart Syndrome: An Unusual Presentation (Soon after Atrial Septal Stenting) of a Relatively Unusual Complication—Experience and Literature Review with an Outlook to Diagnosis and Management

**DOI:** 10.3390/jcm12165357

**Published:** 2023-08-17

**Authors:** Massimiliano Cantinotti, Pietro Marchese, Nadia Assanta, Eliana Franchi, Vitali Pak, Elisa Barberi, Alessandra Pizzuto, Giuseppe Santoro, Raffaele Giordano

**Affiliations:** 1Fondazione G. Monasterio CNR-Regione Toscana, 56124 Pisa, Italy; cantinotti@ftgm.it (M.C.);; 2Adult and Pediatric Cardiac Surgery, Department Advanced Biomedical Sciences, University of Naples “Federico II”, 80138 Napoli, Italy

**Keywords:** thrombosis, hypoplastic left heart syndrome, pediatric cardiology

## Abstract

We started with the experience of thrombus formation in the native aorta of a 3-year-old male child with hypoplastic left heart syndrome (HLHS) and severely hypoplastic but patent mitral and aortic valves after Glenn palliation, which occurred soon after left heart decompression by percutaneous stenting of the atrial septum. The diagnosis was incidental, with the child completely asymptomatic, and progressively subsided in a few days with heparin infusion and chronic warfarin therapy. We reviewed the incidence, diagnosis, and management of native aortic thrombosis in HLHS after different stages of Fontan palliation through a systematic literature search. In all 32 cases, native aortic thrombosis in HLHS was found. The HLHS anatomic subtypes included mitral stenosis/aortic stenosis (fourteen cases or 45.2%), mitral stenosis/aortic atresia (eleven cases or 35.5%), and mitral atresia/aortic atresia (four cases or 12.9%). The age at diagnosis ranged from 13 days to 18 years. Clinical presentation varied from incidental findings, chest pain and/or electrocardiographic abnormalities, cardiac arrest, and transient ischemic attack. Diagnosis was feasible in most of the cases with only transthoracic echocardiography. Mostly (59.4%), patients were treated with anticoagulation, while others underwent surgical (18.7%), direct (12.5%), or systemic (9.3%) thrombolysis. Transplant-free survival was 56.2%, and fatal events occurred in 25%. Major events occurred in 26.3% of those treated with anticoagulation, in 33.3% of patients treated with surgical/systemic thrombolysis, and in 100% of patients treated with direct thrombolysis. In summary, native aortic thrombosis in HLHS may occur at different ages, with a wide spectrum of presentation from incidental finding to a sudden major event. Diagnosis is feasible with transthoracic echocardiography, and management with anticoagulation is effective despite the incidence of major events remaining high.

## 1. Introduction

Native aortic thrombosis in hypoplastic left heart syndrome (HLHS) is a serious and underrecognized complication that may occur at different stages of Fontan palliation and lead to left ventricular (LV) failure, arrhythmias, and death [1,2,3,4,5,6,7,8,9,10,11,12,13,14,15]. Over the past two decades, only small case series of similar complications have been described [2,3,6,7,8,9,13,14,15], while in the past two years, a couple of works [10,11] systematically reviewed single-center experience over Fontan palliation to assess the incidence of such complications.

We herein describe an unusual case of sudden formation of a thrombus in the native aorta of a child with HLHS after an interatrial stenting procedure. We also provide a literature review on the incidence, diagnosis, and management of native aortic thrombosis in HLHS after different stages of Fontan palliation.

## 2. Experience

Data acquisition and analysis were performed in compliance with protocols approved by the Ethical Committee of the Meyer Hospital (ethical approval number 62/2016). Written informed consent was obtained from all participants prior to this study.

A 3-year-old, 15 kg child with HLHS (with severely hypoplastic but patent mitral and aortic valves) and restrictive foramen ovale after the first stage of Nowood–Sano palliation and subsequent Glenn anastomosis at 6 months of age, was admitted for elective percutaneous interatrial stenting. He had previously undergone percutaneous angioplasty for the patent foramen ovale (PFO) at the age of 18 months, as a bridge to the stenting procedure.

The procedure was performed under general anesthesia and echocardiographic guidance. The restrictive PFO was successfully dilated to 10 mm with an Andrastent (26 mm) (Andramed GmbH Schiesswieslenst, Reutlingen, Germany) decreasing the atrial gradient from 8 to 3 mmHg. The baby was weaned from mechanical ventilation in the intensive care unit (ICU) and put under unfractionated heparin (100 UI/kg) and acetylsalicylic acid (5 mg/kg) [3,4]. Then, he was sent to the ward in good clinical conditions and with a mild increase in oxygen saturation (from 80 to 85%). Echocardiography performed the day after the procedure revealed a significant improvement of the transatrial flow. However, despite the lack of clinical findings and a normal electrocardiogram (ECG) (Appendix A), the pre-discharge echocardiography, performed three days after the procedure, imaged a large thrombus within the native aortic root (Figure 1 and Figure 2, Appendix A). Thus, the baby was monitored in ICU and put under low weight heparin (LWE) infusion (25 UI/Kg/h) in view of chronic anticoagulation therapy (warfarin 0.2 mg/Kg) [3,4]. Partial resolution of the clot was documented after 72 h, and it was completed after 6 days of therapy (Figure 3, Appendix A). LWE was stopped after 96 h, and the baby was discharged under warfarin therapy after a few days without clinical and instrumental findings. Of interest, no high-sensitivity (hs) troponin-T rise was noted in serial blood test examinations.

## 3. Literature Review

### 3.1. Methods

In February 2023, we performed a systematic search in the National Library of Medicine for Medical Subject Headings and free text terms including “native aortic thrombus” and “hypoplastic left heart syndrome”. The search was refined by adding keywords for “Fontan palliation” and “univentricular heart”. The cases were excluded if the diagnosis of aortic thrombosis was uncertain or performed post mortem.

### 3.2. Results

From seventeen studies initially selected, two articles were excluded for uncertain diagnosis and one because it was a post mortem analysis. Twelve case reports [1,2,3,5,6,7,8,9,11,12,13,14,15] and two original articles [10,11] met the criteria established and were selected for the analysis. In all 32 cases, native aortic thrombosis in HLHS was described, including our case (Table 1). The HLHS anatomic subtypes included mitral stenosis/aortic stenosis (MS/AS) (fourteen cases or 45.2%), mitral stenosis/aortic atresia (MS/AA) (eleven cases or 35.5%), and mitral atresia/aortic atresia (MA/AA) (four cases or 12.9%). No cases of mitral atresia/aortic stenosis (MA/AS) were described.

Age at diagnosis ranged from 13 days [7] to 18 years [5,10]. Clinical presentation varied from chest pain and/or ECG abnormalities (seventeen cases) [1,2,5,6,10,11,14] to incidental findings (eight cases) [2,7,11], cardiac arrest (three cases) [7,8,10], transient ischemic attack (TIA) (one case) [9], tachycardia (two cases) [7,10], and bradycardia (two cases) [1,12].

Diagnosis was usually made with transthoracic echocardiography [1,2,6,7,9,10,12], while in a few cases, echocardiographic suspicion was confirmed at cardiac catheterization (five cases) [4,5,8,14] or at a computed tomography (CT) scan (three cases) [5,14].

Most of the patients (nineteen cases, e.g., 59.4%) [2,5,7,10,11,14,15] were treated with anticoagulation, surgical thrombolysis was performed in six (18.7%) [4,6,7,8,10], direct thrombolysis in cardiac catheterization theater in four (12.5%) [4,5,7,8], and systemic thrombolysis in three patients [1,12]. Transplant-free survival was 56.2% (e.g., 18 out of 32 cases). Fatal events occurred in eight (25%) cases [4,5,8,14]. Rate of fatal events varied according to different management. Major events occurred in five out of nineteen patients (26.3%) treated with anticoagulation (one transplant, four deaths), in three out six patients (33.3%) who underwent surgical thrombolysis (one death, one transplant) [4,6,7,8,10], and one out of three (33.3%) of those receiving systemic thrombolysis (one death out of three patients) [1,12]. Among the four children treated with direct thrombolysis in cardiac catheterization [4,5,7,8], they all experienced major events (two deaths, two transplants). Thrombus reformation, after complete resolution, occurred in three cases either soon after the resolution or months after [2,7,12].

## 4. Discussion

Native aortic root thrombosis is a rare potential complication in univentricular heart circulation [1,2,3,5,6,7,8,9,10,11,12,13,14,15] that is particularly worrisome since it has the potential for coronary occlusion, resulting in myocardial ischemia and sudden cardiac death, or for systemic embolism [1,2,3,5,6,7,8,9,10,11,12,13,14,15]. Thrombosis in the native aorta has been described only in HLHS with stenotic inflow and/or outflow pathways [1,2,3,5,6,7,8,9,10,11,12,13,14,15]. It may be due to low-flow dynamics inside the left cardiac chambers due to the preferential flow to the right heart chambers in the setting of unrestrictive atrial septal fenestration. In our patient, restrictiveness of the PFO could have forced the blood flow across the left heart pathway, hindering clot formation. At the time of atrial stenting and left heart decompression, the sudden decrease in the blood flow amount and velocity along the left ventricle might have contributed to the thrombosis. The presence of an anterograde flow through a stenotic aortic valve does not seem protective itself [10], rather if the anterograde flow is scarce and at low pressure may compete with the regrade flow creating an area of stasis favoring thrombus formation [10]. We suppose that in our case, before atrial stenting procedure due the stenotic PFO, part of the pulmonary vein flow was forced across the mitral valve and subsequently to the aortic valve (which was of a discrete size for the pathology); thus, the anterograde flow had a pressure enough not to create stasis with the retrograde flow. After the procedure, the flow across the pulmonary veins had a preferential pathway across the PFO; thus, mitral and aortic flow became minimal, favoring clot formation. In this patient, the thrombus was very close to the coronary arteries, and in some images, it may seem to partly obstruct the left coronary artery origin. The risk of myocardial ischemia due either to occlusion or embolization into the vessels was a major concern, although neither an electrocardiograph sign of myocardial ischemia, nor an hs-troponin-T increase was found during continuous monitoring.

In the literature [1,2,3,5,6,7,8,9,10,11,12,13,14,15], this complication has been reported in 32 cases at different ages, from neonates soon after Norwood palliation to adult after the completion of Fontan palliation. Ways of presentation widely varied from incidental findings [2,7,11] in totally asymptomatic children to severe symptoms due to myocardial ischemia, up to sudden death [7,8,10]. The most common presentation was chest pain (accompanied with ECG abnormalities and troponin increase) or arrhythmic events [1,2,5,6,10,11,14].

In our case, the formation of thrombus in the native aorta not only happened with a child completely asymptomatic but also quite unexpectedly after a procedure that theoretically should have improved the hemodynamic balance. The formation of the clot was not related to hypercoagulability due to cardiac catheterization procedure, as one may initially hypothesize, since it was not present the day after the procedure. Rather it was due to a rapid change in the hemodynamics, with a diminished anterograde aortic flow.

Diagnosis is generally feasible with transthoracic echocardiography [1,2,6,7,9,10,11,12], but experience and a careful and systematic assessment of the native aorta are required. The increased awareness of this rare complication has led to an increasing diagnostic rate in the past few years [4,10,11,12], with multiple cases described. In the case we described, the prompt diagnosis at a routine pre-discharge transthoracic echocardiogram and the subsequent therapeutic management made it possible to avoid potentially life-threatening coronary complications. This indicates the importance of systematic echocardiographic investigation of the native aorta as a potential site for thrombus formation in patients with HLHS and hypoplastic inflow and outflow pathways, even when hemodynamic and clinical conditions seem to be optimal [1,2,3,5,6,7,8,9,10,11,12,13,14,15].

Management with anticoagulation [2,5,7,10,11,14] was performed in most cases, while in others, either surgical [6,7,8,9,10], direct [4,5,7,8], or systemic [1,12] thrombolysis was performed. Anticoagulation was effective in ¾ of the cases [2,5,7,10,11,14], while direct thrombolysis [4,5,7,8] always resulted in major events due to distal embolization of fragments of the thrombus. This worrisome complication, which has been rarely reported so far but in the past two years have been repeatedly described [10,11,12], reopen also the debate on the need and the type of anticoagulation required in HLHS. A lifelong anticoagulation therapy seems to be mandatory after such cases, and aspirin alone is not protective [7,11]. In fact, most (if not all) of the children who experienced native aortic thrombosis were under acetylsalicylic acid [7,11], while it may be assumed that none was under warfarin. Of interest, when tested [11], no hypercoagulability defect was found in these patients (including our case) [11].

Even in the case of thrombus resolution, there may be concern about the long-term outcome of such cases. Data on follow-up are scarce [7,10,11], especially those regarding long-term ventricular function [11]. Watking and colleagues [11] reported that at a median follow-up of 8.3 years, four out of six survivors had depressed ventricular function to various degrees. In our case, ventricular function remained unchanged at a 6-month follow-up.

An international registry of similar rare thrombotic complications in univentricular circulation is warranted, since it may help to ameliorate the understanding of the physiopathological triggers; clinical and instrumental findings are to be monitored and to guide in their management in such risky acute conditions.

## Figures and Tables

**Figure 1 jcm-12-05357-f001:**
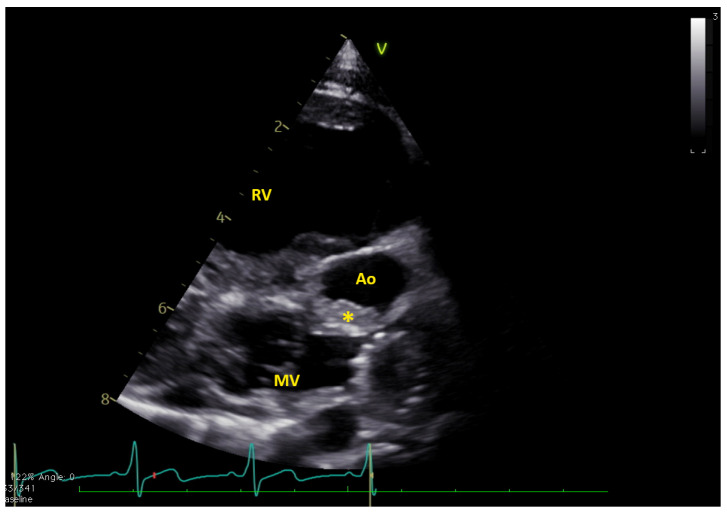
Long axis view showing the thrombus (*) in native aorta (Ao). MV = mitral valve; RV = right ventricle. The native aorta was tricuspid, with moderately hypoplastic annulus (e.g., 7.8 mm, z-score −5.2), while the aortic root was only mildly hypoplastic (13.4 mm, z-score −2.5) (Appendix A).

**Figure 2 jcm-12-05357-f002:**
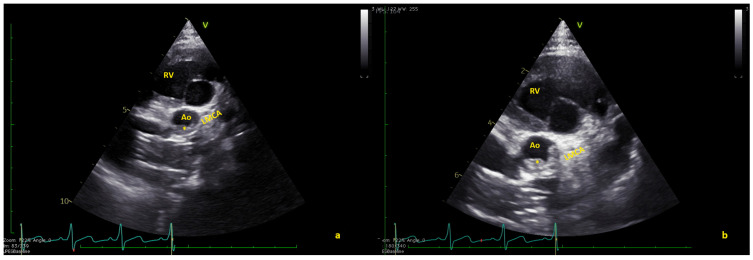
(**a**) Short axis view showing the thrombus (*) in native aorta (Ao). (**b**) In some frames, the thrombus seems to partially obstruct the coronary arteries. LMCA = left main coronary artery; RV = right ventricle (Appendix A).

**Figure 3 jcm-12-05357-f003:**
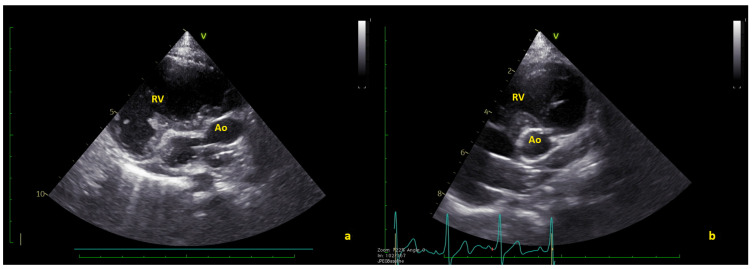
(**a**) Long axis and (**b**) short axis views showing the complete thrombus resolution. Ao = native aorta; RV = right ventricle (Appendix A).

**Table 1 jcm-12-05357-t001:** Studies reporting thrombus in native aorta of HLHS.

Author	Age	Type of HLHS	Stage	Presentation	Diagnosis	Therapy	Alive/ Death	Outcome
Brennan et al., 2001 [9]	3 years	NR	3	TIA, ST changes	TE	Surg THR	Alive	NR
Graham et al., 2006 [6]	5 years	MS/AS	3	ST↓ ↑Tr	Routine TE	Surg THR	Alive	NR
Owens et al., 2006 [2]	6 years	MS/AS	3	Chest pain ↑ST ↑Tr	TE	Direct THR	Death (VF)	
	30 months	MS/AA	2	Incidental finding	Routine TE	Antic.	Alive	Ao thr reformation after Fontan resolved with Antic.
	12 months	MS/AA	2	Incidental finding	Routine TE	Antic.	Alive	SCA after Stage 3
Jansen et al., 2007 [7]	3 weeks	MS/AA	1	Incidental finding	Routine TE	Surg THR	Death (sudden hypotension)	
	1 year	MS/AS	2	Incidental finding	Routine TE	Antic.	Alive	Alive after Stage 3
	2 weeks	MA/AA	1	NS VT	TE	THR	Alive	Ao thr reformation at 4 months
	13 days	MS/AS	1	CA	TE	Antic	Alive	Alive after Stage 2
Mokerijee et al., 2007 [15]	8 months	MS/AA	2	Ischemic ECG	Cath	Antic.	Alive	Alive
Mitchell et al., 2015 [8]	13 months	MS/AS	2	Syncope, heart block	Cath	Direct THR° followed by Surg THR	Alive	Transp.
	22 days	MS/AA	1	Failure to weaning from MV	Cath	Surg THR	Death (stroke)	
Keraliya et al., 2016 [14]	22 years	NR	3	Chest pain, T inversion	CT	Antic.	Alive	NR
Rajab et al., 2020 [10]	39 months	MS/AS	2	Cyanosis, SVT	Cath	Direct THR	Alive	Transp.
	26 months	MS/AA	2	VT arrest	TE	Surg THR	Alive	Transp.
	5 months	MS/AS	2	Wide QRS tach	TE	Surg THR	Transp.	Transp.
	18 years	MS/AS	3	Chest pain ↑Tr	TE	Antic.	Alive, persistent Ao thr	Not suitable for transp.
Patel et al., 2020 [5]	18 years	MS/AA	3	Chest pain Breathless ST↓, ↑Tr	Cath	Direct THR	death (Probably stroke)	
	17 years	MA/AA	3	Chest pain ST↓, ↑Tr	CT	Antic. (heparin, plavix, aspirin)	CA (stroke)	Alive
	13 years	MS/AA	3	Abdominal pain, 1st degree AV block, chest pain, ST↓, ↑Tr	CT	NR	Alive	List for transplant
Segar et al., 2021 [5]	3 years	MS/AA	2	Bradycardia, ↑Tr	TE	Systemic THR	ECMO, Ao thr reformation Death	
Abraham et al., 2021 [1]	15 days	MS/AS	1	Bradycardia, hypotension, ↑ST	TE	Systemic THR	Alive	Follow-up after Glenn
Watkins et al., 2022 [11]	2.1 (0.3–3) years	4 MS/AS 3 MS/AA 2 MA/AA	Stage 1: 2 Stage 2: 3 Stage 3: 4	Six with symptoms (chest pain, heart failure) three with incidental findings	TE (1 confirmed at Cath)	Antic. Seven with heparin, two with aspirin	Three deaths one CA, one cyanosis and CA, one stroke	Follow-up at 8.3 years (IQR 1.3–10.8 years) All alive

Ao = aorta; Antic. = anticoagulation, AA = aortic atresia, AS = aortic stenosis, Ao thr = aortic thrombus, AV = atrio-ventricular, CT = computed tomography, CA = cardiac arrest, Cath = catheterization, ECG = electrocardiogram, ECMO = extracorporeal membrane oxygenation, MA = aortic atresia, MS = mitral stenosis, MV = mechanical ventilation, NSVT = nonsustained ventricular tachycardia; Surg THR = surgical thrombectomy, SCA = sudden cardiac death, TE = thoracic echocardiography, TIA = transient ischemic attack, THR = direct thrombolysis, thr = thrombosis, Tr. = troponin, SVT = supraventricular tachycardia, VT = ventricular tachycardia, Transp. = transplant; ° the procedure was performed twice with embolization into the first diagonal and right coronary artery without ST changes, hemodynamic changes or arrhythmias.

## Data Availability

The data presented in this study are available on request from the corresponding author.

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
