# Peer review of "Native Aortic Root Thrombosis in Hypoplastic Left Heart Syndrome: An Unusual Presentation (Soon after Atrial Septal Stenting) of a Relatively Unusual Complication—Experience and Literature Review with an Outlook to Diagnosis and Management"

_jcm, 2023, doi:10.3390/jcm12165357_

Round 1

Reviewer 1 Report

Great job on presentation of this rare complication and extensive review of literature. I enjoyed reading this case and congratulate the authors on the favorable outcome of their patient. 

1. The description of the patient appears to be a 3 year old HLHS s/p Norwood Sano. It would be helpful to know why the patient did not go forward with stage 2 and 3 palliation and is still only post Norwood-Sano operation. Is the patient considered high risk? Was there reduced right ventricular function? Was there anything in this patient's history preventing further operations, that would otherwise explain this?

2. I am curious to know if this patient underwent a hypercoagulable workup.

3. It would be nice to attach a copy of the patients EKG. 

1. Lines 11-13 of the discussion have incorrect grammar and are difficult to understand. 

2. Sporadic incorrect use of articles and prepositions.

Author Response

Dear review many thanks for you suggestion

Great job on presentation of this rare complication and extensive review of literature. I enjoyed reading this case and congratulate the authors on the favorable outcome of their patient. 

  1. The description of the patient appears to be a 3 year old HLHS s/p Norwood Sano. It would be helpful to know why the patient did not go forward with stage 2 and 3 palliation and is still only post Norwood-Sano operation. Is the patient considered high risk? Was there reduced right ventricular function? Was there anything in this patient's history preventing further operations, that would otherwise explain this?

Response_ We apologize for the mistake; the baby underwent previous Glenn anastomosis. We have corrected in the text.

  1. I am curious to know if this patient underwent a hypercoagulable workup.

Response: we have done and was negative, we added in the discussion

  1. It would be nice to attach a copy of the patients EKG.

Response: we attached and EKG as requested by the reviewer 

Comments on the Quality of English Language

  1. Lines 11-13 of the discussion have incorrect grammar and are difficult to understand. 

Response: we edit this paragraph. We apologize for the misunderstanding.

  1. Sporadic incorrect use of articles and prepositions. Response: we corrected

Reviewer 2 Report

The authors reported case of native aortic root thrombosis in HLHS soon after septal stenting. Furthermore, the authors performed a literature review.

Although this paper had been clearly reported about native aortic root thrombosis in HLHS, I found it a little difficult to decide what to make of this report.

Abstract:

Well written and clear

Introduction:

Well written and clear

Case Report:

1.     This case is a 3 years-old, who underwent first stage of Norwood-Sano palliation. However, he has not been able to reach Glenn procedure and Fontan procedure. He had a restrictive foramen ovale. Could that be the cause? Does he have other hemodynamic issues? Were there any laboratory examination problems, such as polycythemia? There may be a problem with hemodynamics and laboratory examination, but there is no mention of that.

Literature review:

Well written and clear

Discussion:

1.     The Authors described that the mechanism of thrombus formation is considered to low-flow from inside left cardiac chambers due to preferential flow to right heart chamber. However. What about other factors?  No description of the aortic morphology such as the length of aortic root and aortic valve size. Native aortic root morphology may also be involved in thrombus formation.

2.     The Authors described that anticoagulation was effective and direct thrombolysis resulted in major events. Concerning prognosis, the degree of thrombosis spread (within the aorta or within the coronary), symptoms, and ECG abnormalities may be involved. In this case, there was thrombus only in the aorta. Furthermore, there was no symptom and a normal ECG. Therefore, it may have followed a good course.    

This case report is about thrombotic complication in HLHS and importance of monitoring and management. However, Similar reports have been published in the past, so this report lacks novelty.

Author Response

Dear Editor many thanks for your suggestions.

The authors reported case of native aortic root thrombosis in HLHS soon after septal stenting. Furthermore, the authors performed a literature review.

Although this paper had been clearly reported about native aortic root thrombosis in HLHS, I found it a little difficult to decide what to make of this report.

Abstract:

Well written and clear

Introduction: 

Well written and clear

Case Report:

  1. This case is a 3 years-old, who underwent first stage of Norwood-Sano palliation. However, he has not been able to reach Glenn procedure and Fontan procedure. He had a restrictive foramen ovale. Could that be the cause? Does he have other hemodynamic issues? Were there any laboratory examination problems, such as polycythemia? There may be a problem with hemodynamics and laboratory examination, but there is no mention of that.

Response: We apologize for the mistake; the baby underwent previous Glenn anastomosis. We have corrected in the text.

  1.  

Literature review:

Well written and clear

Discussion:

  1. The Authors described that the mechanism of thrombus formation is considered to low-flow from inside left cardiac chambers due to preferential flow to right heart chamber. However. What about other factors? No description of the aortic morphology such as the length of aortic root and aortic valve size. Native aortic root morphology may also be involved in thrombus formation.

Response: we added information about the size of the aortic valve in Figure 1. We report measurements and z-scores. We also add a comment on aortic size in the discussion.

  1. The Authors described that anticoagulation was effective and direct thrombolysis resulted in major events. Concerning prognosis, the degree of thrombosis spread (within the aorta or within the coronary), symptoms, and ECG abnormalities may be involved. In this case, there was thrombus only in the aorta. Furthermore, there was no symptom and a normal ECG. Therefore, it may have followed a good course.   

Rsponse: we have described that the thrombus was very close to the coronary artery. We have tried to emphasize better in the discussion section that coronary artery occlusion (due to the thrombus itself or its embolization) was a major concern.

This case report is about thrombotic complication in HLHS and importance of monitoring and management. However, Similar reports have been published in the past, so this report lacks novelty.

Response: the case we have described is different from previous one. This complication has been curiously described almost only in the last few years, so we retain that has been not sufficient awareness so far. Furthermore we performed un updated literature review on the incidence, signs and symptoms, methods for diagnosis,  and therapeutical strategies that may orientate in case of a similar complication.

Round 2

Reviewer 2 Report

The authors have faithfully corrected the issues I pointed out.

As the authors state, thrombotic complication in HLHS is serious problem. And I believe that monitoring and management are important. Similar reports have been published in the past. However, the Authors state an updated literature review. Therefore, I think that this review has important message.

Comment:

1:Check the Z-score of the hypoplastic annulus diameter of the native aorta

Author Response

Comments and Suggestions for Authors

The authors have faithfully corrected the issues I pointed out.

As the authors state, thrombotic complication in HLHS is serious problem. And I believe that monitoring and management are important. Similar reports have been published in the past. However, the Authors state an updated literature review. Therefore, I think that this review has important message.

Response: Thanks for your revision and consideration.

Comment: 

1:Check the Z-score of the hypoplastic annulus diameter of the native aorta

Response: We performed.